https://doi.org/10.1038/s41467-020-17182-9　　**OPEN**

# Two-dimensional optomechanical crystal cavity with high quantum cooperativity

Hengjiang Ren[1,2,3], Matthew H. Matheny[1,2,3], Gregory S. MacCabe[1,2,3], Jie Luo[1,2,3], Hannes Pfeifer [4,5], Mohammad Mirhosseini[1,2,3] & Oskar Painter [1,2,3✉]

Optomechanical systems offer new opportunities in quantum information processing and quantum sensing. Many solid-state quantum devices operate at millikelvin temperatures—however, it has proven challenging to operate nanoscale optomechanical devices at these ultralow temperatures due to their limited thermal conductance and parasitic optical absorption. Here, we present a two-dimensional optomechanical crystal resonator capable of achieving large cooperativity $C$ and small effective bath occupancy $n_b$, resulting in a quantum cooperativity $C_{eff} \equiv C/n_b > 1$ under continuous-wave optical driving. This is realized using a two-dimensional phononic bandgap structure to host the optomechanical cavity, simultaneously isolating the acoustic mode of interest in the bandgap while allowing heat to be removed by phonon modes outside of the bandgap. This achievement paves the way for a variety of applications requiring quantum-coherent optomechanical interactions, such as transducers capable of bi-directional conversion of quantum states between microwave frequency superconducting quantum circuits and optical photons in a fiber optic network.

[1] Thomas J. Watson, Sr., Laboratory of Applied Physics, California Institute of Technology, Pasadena, CA 91125, USA. [2] Kavli Nanoscience Institute, California Institute of Technology, Pasadena, CA 91125, USA. [3] Institute for Quantum Information and Matter, California Institute of Technology, Pasadena, CA 91125, USA. [4] Max Planck Institute for the Science of Light, Staudtstrasse 2, 91058 Erlangen, Germany. [5] Present address: Institut für Angewandte Physik, Universität Bonn, Wegelerstraße 8, 53115 Bonn, Germany. ✉email: opainter@caltech.edu

Recent advances in optomechanical systems, in which mechanical resonators are coupled to electromagnetic waveguides and cavities[1,2], have led to a series of scientific and technical advances in areas such as precision sensing[3,4], nonlinear optics[5,6], nonreciprocal devices[7–9], and topological wave phenomena[10,11]. In addition, such systems have been used to explore macroscopic quantum phenomena, from initial demonstrations of laser cooling of mechanical resonators into their quantum ground state[12–16] to heralded preparation and entanglement of mechanical quantum states[17–20], generation of squeezed light[6,21], and coherent transduction between photons with different energies[5,22–26].

Optomechanical crystals (OMCs)[27], where electromagnetic and elastic waves overlap within a lattice, are patterned structures that can be engineered to yield large radiation–pressure coupling between cavity photons and phonons. Previous work has realized one-dimensional (1D) silicon (Si) OMC cavities with extremely large vacuum optomechanical coupling rates ($g_0 \approx 1$ MHz)[28,29], enabling a variety of applications in quantum optomechanics including the aforementioned ground-state cooling[13] and remote quantum entanglement of mechanical oscillators via an optical channel[19]. An application area of growing interest for OMCs is in hybrid quantum systems involving microwave-frequency super-conducting quantum circuits[30,31]. Owing to the large ratio ($\times 10^5$) of the speed of light to the speed of sound in most materials, OMCs operating at telecom-band optical frequency naturally couple strongly to similar wavelength microwave-frequency acoustic modes. Recent experimental demonstrations of microwave-frequency phononic crystal cavities with ultralow dissipation[32] and strong dispersive coupling to superconducting qubits[33] indicate that there are potentially significant technical advantages in forming an integrated quantum electrodynamic and acoustodynamic circuit architecture for quantum information processing[34,35]. In such an architecture, OMCs could provide a quantum interface between microwave-frequency logic circuits and optical quantum communication channels.

A significant roadblock to further application of OMC cavities for quantum applications is the very weak, yet non-negligible parasitic optical absorption in current devices[17–20,36]. Optical absorption, thought to occur due to surface defect states[37,38], together with inefficient thermalization due to the 1D nature of Si nanobeam OMC cavities currently in use, can yield significant heating of the microwave-frequency acoustic mode of the device. At ultralow temperatures ($\lesssim 0.1$ K), where microwave-frequency systems can be reliably operated as quantum devices, optical absorption leads to rapid (sub-microsecond) heating of the acoustic cavity mode[36]. This has limited quantum optomechanical experiments to schemes with high optical power and short pulses[17–20,32,36] or very low continuous optical power[39,40].

The most relevant figure of merit for quantum optomechanical applications, when there is thermal population in the system, is the effective quantum cooperativity ($C_{\mathrm{eff}} \equiv C/n_{\mathrm{b}}$), corresponding to the standard photon–phonon cooperativity ($C \equiv \gamma_{\mathrm{OM}}/\gamma_{\mathrm{b}}$) divided by the Bose factor of the effective thermal bath ($n_{\mathrm{b}}$) coupled to the acoustic mode of the cavity[5,23,36]. Here $\gamma_{\mathrm{b}}$ represents the total coupling rate of the mechanical system to its various thermal baths. In previous experiments with nanobeam OMC cavities at millikelvin temperatures, the quantum cooperativity was substantially degraded owing to the heating and damping caused by the optical-absorption-induced hot bath. The heating of the acoustic cavity mode by the optically generated hot bath can be mitigated through several different methods. The simplest approach in a low temperature environment is to couple the cavity more strongly to the surrounding cold bath of the chip or through addition of another cold bath as in experiments in a $^3$He buffer gas environment[41,42]. This method can be quite effective in decreasing the acoustic mode thermal occupancy in the presence of optical absorption; however, the effectiveness of the method relies on increasing the coupling to baths other than the optical channel, which necessarily decreases the overall photon–phonon quantum cooperativity.

Here we employ a strategy that makes use of the frequency-dependent density of phonon states within a phononic bandgap structure to overcome this limitation. Using a two-dimensional (2D) OMC cavity[43–45], the thermal conductance between the hot bath and the cold environment is greatly increased owing to the larger contact area of the 2D structure with the bath, while the acoustic mode of interest is kept isolated from the environment through the phononic bandgap of the structure. By keeping the intrinsic damping of acoustic mode low, this method is a promising route to realizing $C_{\mathrm{eff}} > 1$. Initial work in this direction, performed at room temperature, utilized snowflake-shaped holes in a Si membrane to create a quasi-2D OMC with substantially higher optical power handling capability, although with a relatively low optomechanical coupling of $g_0/2\pi = 220$ kHz[45]. In this work, we realize a Si quasi-2D OMC with over 50-fold improvement in optomechanical back-action per photon and a much higher thermal conductance ($\times 68$) compared to 1D structures at millikelvin temperatures. Most importantly, we demonstrate a $Q$-factor of $1.2^{+0.12}_{-0.15} \times 10^9$ for the 10-GHz optomechanically coupled acoustic mode of the cavity and a $C_{\mathrm{eff}}$ greater than unity under continuous-wave optical pumping, suitable for realizing applications such as signal transduction of itinerant quantum signals[22–25].

## Results

**Design of the quasi-2D OMC cavity.** The quasi-2D OMC cavity in this work is designed around the silicon-on-insulator (SOI) materials platform, which naturally provides for a thin Si device layer of a few hundred nanometers in which both microwave-frequency acoustic modes and near-infrared optical modes can be guided in the vertical direction[46]. Patterning of the Si slab through plasma etching is used to form a nanoscale lattice supporting Bloch waves for both optical and acoustic modes. We focus on the fundamental guided optical of even vector parity about the center of the Si slab ($\sigma_z = +1$). This choice is motivated by the fact that for a connected lattice of low air-filling fraction the fundamental $\sigma_z = +1$ optical modes are the most strongly guided in the Si slab, greatly reducing their sensitivity to scattering loss. It is common to refer to these modes as transverse-electric (TE-like), as their electric field polarization lies predominantly in the plane of the slab. For a symmetric Si slab, only the acoustic modes of $\sigma_z = +1$ are coupled via radiation pressure to the optical modes of the slab. The OMC cavity design consists of three major steps.

First, we start with a periodically patterned quasi-2D slab structure with both phononic and photonic bandgaps in which to host the optomechanical cavity. Here we use the "snowflake" crystal with a hexagonal lattice[46] as shown in Fig. 1a. The snowflake crystal provides a pseudo-bandgap for TE-like optical-guided waves and a full bandgap for all acoustic mode polarizations. Finite-element-method (FEM) simulations of the optical and acoustic modes of the snowflake crystal were performed using the COMSOL software package[47], with nominal snowflake parameters corresponding to a Si slab thickness of $t = 220$ nm and $(a, r, w) = (500, 205, 75)$ nm, resulting in a TE-like guided-mode photonic bandgap extending over optical frequencies of 180–240 THz (vacuum wavelength 1250–1667 nm) and an acoustic bandgap covering 8.85–11.05 GHz.

Second, we create an in-plane waveguide in the snowflake lattice. This is done by replacing one row of snowflake unit cells

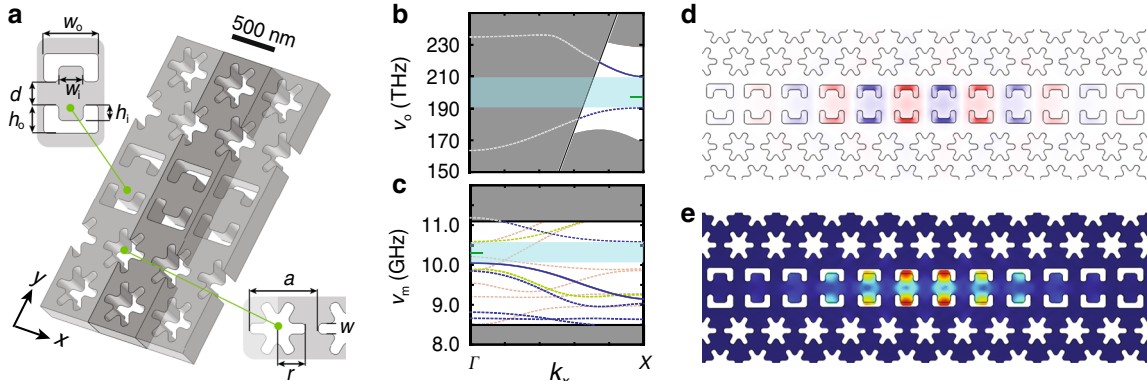

**Fig. 1 Quasi-2D OMC cavity design. a** Unit cell schematic of a linear waveguide formed in the snowflake crystal. Guided modes of the waveguide propagate along the x-axis. Insets: (left) "C"-shaped parameters; (right) snowflake parameters. **b** Photonic and **c** phononic bandstructure of the linear waveguide. The solid blue curves are waveguide bands of interest; dashed lines are other guided modes; shaded light blue regions are bandgaps of interest; green tick mark indicates the cavity mode frequencies; gray regions denote the continua of propagating modes. In the photonic bandstructure, only modes of even vector parity about the center of the Si slab ($\sigma_z = +1$) are shown. In the acoustic bandstructure, green dashed curves are for $\sigma_z = +1$ and $\sigma_y = -1$ parity modes, and yellow dashed curves denote $\sigma_z = -1$ modes. **d** FEM-simulated mode profile ($E_y$ component of the electric field) of the fundamental optical cavity resonance at $\omega_c/2\pi = 194$ THz, with red (blue) corresponding to positive (negative) field amplitude. **e** Simulated displacement profile of the fundamental acoustic cavity resonance at $\omega_c/2\pi = 10.27$ GHz. The magnitude of displacement is represented by color (large displacement in red, zero displacement in blue).

with a customized unit cell. Waveguiding to this line-defect occurs for photon and phonon modes that lie within the corresponding bandgaps of the surrounding snowflake lattice. Here we chose to form the line-defect by replacing one row of snowflakes with a set of "C"-shaped holes. This design is inspired by the 1D nanobeam OMCs reported in ref. [28]. Optomechanical coupling in this sort of design is a result of both bulk (photoelastic)[48] and surface (moving boundary)[49] effects. The "C" shape allows for large overlap of the acoustic mode stress field with the optical mode intensity in the bulk of the Si device layer, while also focusing the optical mode at the air–Si boundary to increase the moving boundary contribution to the optomechanical coupling. The width of the line-defect and exact shape and dimension of the "C" shape were optimized considering several factors: (i) large guided-mode vacuum coupling rate $g_\Delta$[46], (ii) avoidance of leaky optical resonances of the slab, and (iii) creation of guided acoustic bands with dispersion. Leaky optical resonances are resonant with the Si slab yet lie above the light cone; imperfections in the fabricated structure can result in large coupling between the guided optical mode of interest and leaky resonances at the same frequency, resulting in large scattering loss. Acoustic bands of limited dispersion (flat bands) are also susceptible to fabrication imperfections as these acoustic modes tends to localize around small defects resulting in poor overlap with the more extended optical modes (this was a primary difficulty in prior 2D snowflake OMC work[6]). Photonic and phononic bandstructure diagrams of the optimized waveguide unit cell are shown in Fig. 1b, c, respectively. Shaded in light blue are the optical guided-mode bandgap extending from 190 to 210 THz (vacuum wavelength 1430–1580 nm) and the acoustic guided-mode bandgap extending from 10 to 10.6 GHz. Abutting these bandgaps and plotted as solid blue curves are the optical and acoustic waveguide bands of interest.

The final step in the cavity design involves introducing a tapering of the line-defect waveguide properties along the waveguide propagation direction (x-axis). Here we utilize a modulation of the "C"-shaped parameters that increases quadratically in amplitude with distance along the x-axis of the line-defect waveguide from a designated center position of the cavity. This introduces an approximate quadratic shift of the frequency of the waveguide modes with distance from the cavity center.

For waveguide modes near a band edge, this results in localization of the modes as they are pushed into a bandgap away from the cavity center. As detailed in Supplementary Note 1, a Nelder–Mead simplex search algorithm was used to obtain a tapered cavity structure with simultaneously high optical Q-factor and large optomechanical coupling between co-localized optical and acoustic modes. Figure 1d, e displays the resulting simulated field profiles of the fundamental optical resonance ($\omega_c/2\pi = 194$ THz, $\lambda_c = 1550$ nm) and coupled acoustic resonance ($\omega_m/2\pi = 10.27$ GHz) of the optimized 2D OMC cavity, respectively. The co-localized modes have a theoretical vacuum optomechanical coupling rate of $g_0/2\pi = 1.4$ MHz, and the optical mode has a theoretical scattering-limited quality factor of $Q_{c,\text{scat}} = 2.1 \times 10^7$.

Test devices based on this new design were fabricated from a SOI microchip with a 220-nm-thick Si device layer and an underlying 3 μm buried oxide layer. A scanning electron micrograph (SEM) image of a fabricated 2D OMC cavity and optical waveguide for coupling light into the structure are shown in Fig. 2a, b. Several iterations of fabrication were performed in order to improve the fidelity of the fabrication with respect to the design structure. Between fabrication iterations, SEM images of devices were analyzed to determine the fabricated geometrical parameters of the cavity structure; this information was fed back into the next fabrication iteration in order to realize devices with a geometry as close as possible to the simulation-optimized design parameters. An example of fitted "C"-shaped and snowflake holes are shown as red solid lines in the SEM image of Fig. 2b, with corresponding fitted cavity parameters plotted in Fig. 2c.

**Optomechanical coupling and mechanical damping.** Fabricated devices were characterized both at room temperature (300 K) and at cryogenic temperatures inside a fridge ($T_f = 10$ mK). A simplified schematic of the optical measurement set-up is shown in Fig. 2d. Room temperature testing was performed using a dimpled optical fiber taper to evanescently couple light into and out of a chip-based Si coupling waveguide[50]; each Si coupling waveguide is butt-coupled to a corresponding OMC cavity as shown in Fig. 2a (also see Supplementary Note 3). A typical optical spectrum from one of the quasi-2D OMC cavities is

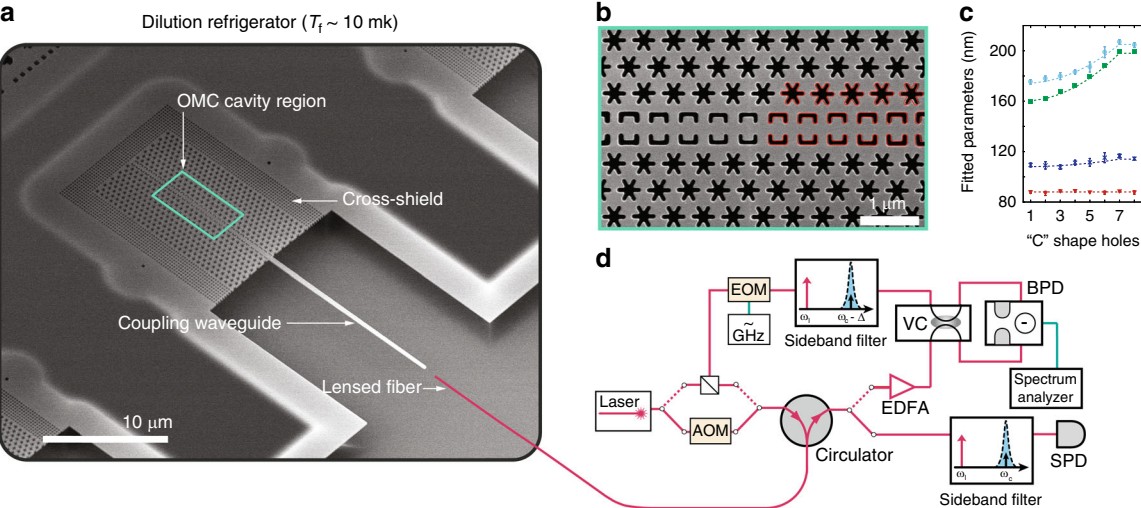

**Fig. 2 Device fabrication and measurement set-up. a** SEM image of a full quasi-2D snowflake OMC device fabricated on SOI. This device is an "8-shield device" in which an additional eight periods of cross-structure phononic bandgap shielding is applied at the periphery (see Supplementary Note 4). A lensed optical fiber is used to couple light into a tapered on-chip waveguide, which is butt-coupling to the quasi-2D snowflake OMC cavity. **b** SEM image of the center of the cavity region. Fitting to the geometries for "C"-shaped holes and snowflake holes in the cavity region are shown as red solid lines. **c** Measured "C" parameters fit from SEM images of a fabricated cavity (dots) along with their design values (dashed curves). Shown over one half of the cavity (first 8 "C" shapes on the right side of the cavity) are the parameters (from top to bottom): $h_o$ (cyan circles), $w_o/2$ (green squares), $h_i$ (blue crosses), and $w_i/2$ (red triangles). Error bars in geometrical parameters correspond to standard deviation of ten sets of measured SEM images. **d** Experimental set-up for characterization of the quasi-2D snowflake OMC cavity. Multiple optical switches are used to switch between continuous wave and pulsed optical excitation and between heterodyne spectroscopy and single photon detection. AOM acousto-optic modulator, EOM electro-optic modulator, EDFA erbium-doped fiber amplifier, VC variable coupler, BDP balanced photodetector, SPD single photon detector. A more detailed schematic and description of the measurement set-up is provided in Supplementary Note 2.

displayed in Fig. 3a, showing a fundamental optical resonance at a wavelength of $\lambda_c = 1558.8$ nm with a loaded (intrinsic) optical $Q$-factor of $Q_c = 3.9 \times 10^5$ ($Q_{c,i} = 5.3 \times 10^5$).

In order to measure the coupled acoustic resonance(s) of the OMC cavity, we used a pump–probe "electromagnetically induced transparency" measurement[51,52] (see Supplementary Note 5). Coherent detection of the beating of the pump and probe laser tones with heterodyne spectroscopy produces a spectrum of the coupled acoustic modes. For the new quasi-2D OMC cavity design, we found a single, dominantly coupled acoustic mode around $\omega_m/2\pi \approx 10.2$ GHz. A plot of the measured acoustic mode spectrum at several optical pump powers is shown in Fig. 3b for the device of Fig. 2a. Optomechanical back-action from the pump laser can be seen to broaden the acoustic resonance; a plot of the fit resonance linewidth ($\gamma = \gamma_0 + \gamma_\phi + \gamma_P + \gamma_{OM}$) versus intra-cavity photon number of the pump laser tone ($n_c$) is shown as an inset to Fig. 3b. Here $\gamma_0$ is the intrinsic energy decay rate, $\gamma_P$ is the optical absorption bath-induced damping, $\gamma_\phi$ is due to any pure dephasing effects (frequency jitter) of the acoustic resonance, and $\gamma_{OM} = 4g_0^2 n_c/\kappa$ is the optomechanical back-action rate[32]. From the slope of the back-action-broadened linewidth versus $n_c$, we extract a vacuum coupling rate of $g_0/2\pi = 1.09^{+0.13}_{-0.09}$ MHz, close to the simulated optimum value of 1.4 MHz.

Following initial room temperature measurements, the new optimized quasi-2D OMC cavities were tested at millikelvin temperatures. We measured the intrinsic mechanical damping rate, back-action cooling, and heating dynamics of the OMC cavities in a dilution refrigerator (DR) with a temperature of $T_f \approx$ 10 mK at the base plate connected to the mixing chamber. The 1-cm square sample containing an array of devices was directly mounted on a copper mount attached to the mixing chamber plate, and a 3-axis stage was used to align a lensed optical fiber to the tapered on-chip coupling waveguide of a given device under test (see Fig. 2a and Supplementary Note 3).

We measured the intrinsic mechanical $Q$-factor of the quasi-2D OMC devices at millikelvin temperatures using an pulsed optical scheme in which 10-$\mu$s-long optical pulses excite and read-out the energy in the acoustic mode (for details, see the "Methods" section). By varying the delay between the optical pulses, this technique allows for the evaluation of the acoustic energy ringdown while the laser light field is off[32,53]. In Fig. 3c, we show the measured ringdown curve for a 10-GHz acoustic mode of a quasi-2D snowflake OMC cavity with an additional acoustic shield. In order to increase the acoustic isolation of the acoustic cavity mode, on this device we added a periphery consisting of eight periods of a cross-structure phononic bandgap shield (see Supplementary Note 4). Fitting of the initial mode occupancy at the beginning of an optical pulse ($n_i$) versus inter-pulse delay ($\tau_{off}$) yields an intrinsic acoustic energy decay rate of $\gamma_0/2\pi = 8.28^{+1.25}_{-0.43}$ Hz, corresponding to a mechanical $Q$-factor of $Q_m = 1.2^{+0.12}_{-0.15} \times 10^9$. This large mechanical $Q$-factor is consistent with other measurements of Si nanobeam OMC cavities at millikelvin temperatures[32] and is thought to result from a suppression of acoustic absorption from near-resonant two-level system defects[54].

**Optical-absorption-induced hot bath.** While the acoustic mode $Q$-factor measured in ringdown measurements is promising for certain quantum memory applications[34,35], it is measured with the laser pump off. The prospects for performing coherent quantum operations between photons and phonons depends critically on the ability to minimize unwanted heating and damping of the acoustic mode due to parasitic effects resulting from optical absorption in the presence of an applied laser field. A model for the heating and damping in Si OMC cavities, first proposed in ref. [39], is illustrated in Fig. 4a. In this model, the acoustic mode of the OMC cavity is weakly coupled to the surrounding DR environment at a rate $\gamma_0$, while simultaneously

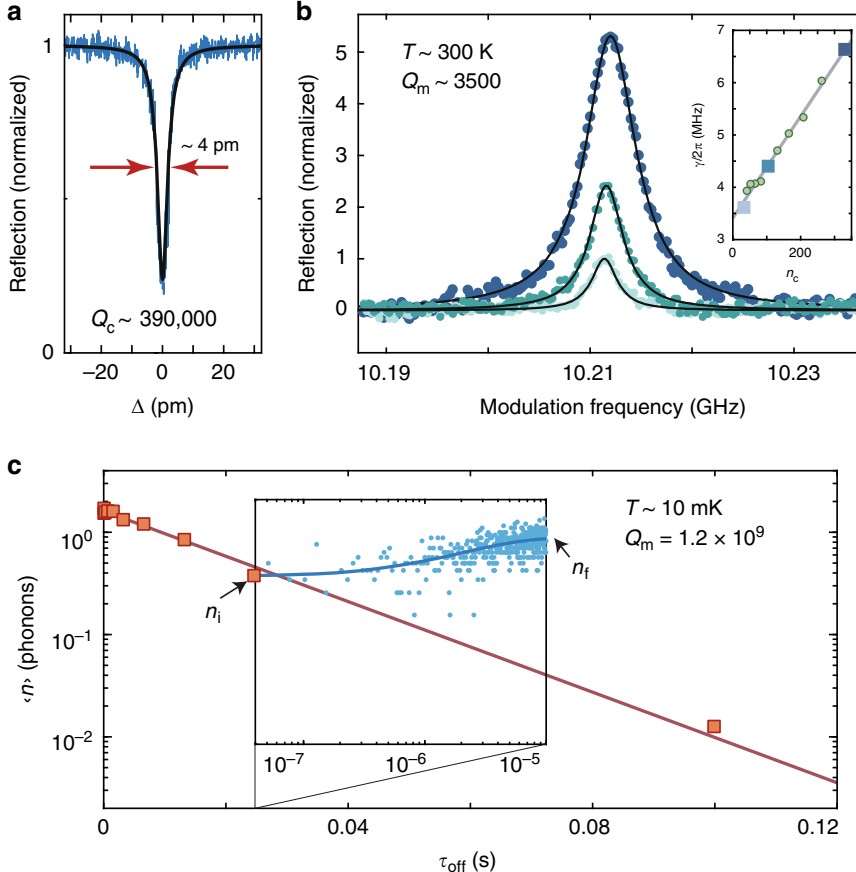

**Fig. 3 Optomechanical characterization. a** Optical spectrum of a quasi-2D snowflake OMC cavity measured using a swept laser scan. **b** Pump–probe measurement of the mechanical mode spectrum of interest centered around $\omega_m/2\pi = 10.21$ GHz for different optical pump powers (from lowest to highest peak reflection: $n_c = 33$, $n_c = 104$, and $n_c = 330$). Inset: measured mechanical mode linewidth versus $n_c$. The device measured in **a**, **b** is a cavity with zero additional acoustic shield periods (zero-shield device). **c** Pulsed ringdown measurement of a quasi-2D snowflake OMC cavity with 8 periods of additional cross-structure shielding (8-shield device). The measured optomechanical parameters of this device are $(\kappa, \kappa_e, g_0, \omega_m, \gamma_0) = 2\pi(1.19$ GHz, 180 MHz, 1.18 MHz, 10.02 GHz, $8.28^{+1.25}_{-0.43}$ Hz). The phonon occupancy at the beginning of each 10-µs optical pulse, $n_i$ (orange squares), is measured using a peak power corresponding to $n_c = 60$. Inset: mode heating within the optical pulse, with blue circles corresponding to data and the solid line a fit to the heating curve. $n_f$ is the mode occupancy at the end of the optical pulse, used here to check for consistent excitation across the different inter-pulse $\tau_{off}$ measurements.

being coupled to an optically generated hot bath at a rate $\gamma_p$. The source of the hot bath in Si OMC devices is thought to be due to linear optical absorption via electronic defect states at the surface of the etched Si[37,38], which through phonon-mediated relaxation processes produces a bath of hot phonons that pile up above the acoustic bandgap of the OMC structure due to phonon bottleneck effects[32]. Assuming a phonon density of states corresponding to that of a 2D plate for the hot bath above the OMC bandgap, and weak coupling to the localized acoustic cavity mode via 3-phonon scattering, such a model predicts heating of the localized acoustic mode that scales as $n_c^{1/3}$ and damping that scales as $n_c^{2/3}$.

Here we explore the optically induced parasitic heating and damping for the quasi-2D OMC cavity. As for the previous 1D nanobeam measurements, optical measurements were performed using a continuous-wave pump laser tuned to the optical cavity resonance ($\Delta = 0$) to avoid optomechanical back-action cooling and damping (see "Methods"). Results of the optically induced heating and damping of the 10 GHz breathing-like mode of the 2D OMC cavity are displayed in Fig. 4b, c, respectively. In Fig. 4b, the inferred hot bath occupancy $n_p$ (left vertical axis) and the corresponding bath temperature $T_p$ (right vertical axis) are plotted versus intra-cavity photon number $n_c$ (lower horizontal axis) and the corresponding input power (upper horizontal axis).

We find that the hot bath occupancy is accurately fit by the power law, $n_p = (1.1) \times n_c^{0.3}$. This is the same power-law scaling as found for 1D nanobeam OMCs in ref. [32] (red dashed line in Fig. 4b); however, the overall magnitude of the hot bath occupancy has substantially dropped for the quasi-2D cavity by a factor of 7.2. Using a modified thermal conductance model that assumes ballistic phonon transport and a power-law dependence on temperature consistent with our measurements, $C_{th} = \epsilon(T_p)^{\alpha=2.3}$, FEM numerical simulations of the 1D and 2D cavity structures (see Supplementary Note 6) predict a greatly enhanced thermal conductance coefficient for the quasi-2D cavity $\epsilon_{2D}/\epsilon_{1D} = 42$. This yields a hot bath occupancy ratio between nanobeam and quasi-2D cavities of 6.2, in good correspondence to the measured value. In alignment with our design strategy, the reduction in mode heating according to this model is primarily the result of a geometric effect due to the fact that the quasi-2D cavity is connected to the surrounding chip structure over a larger in-plane solid angle than that of the 1D nanobeam and thus has a larger number of phonon modes to carry heat away (i.e., a larger thermal conductance).

In Fig. 4c, we plot the spectral linewidth of the breathing mode of the quasi-2D cavity, determined in this case by measuring the acoustic mode thermal noise spectrum imprinted on the reflected

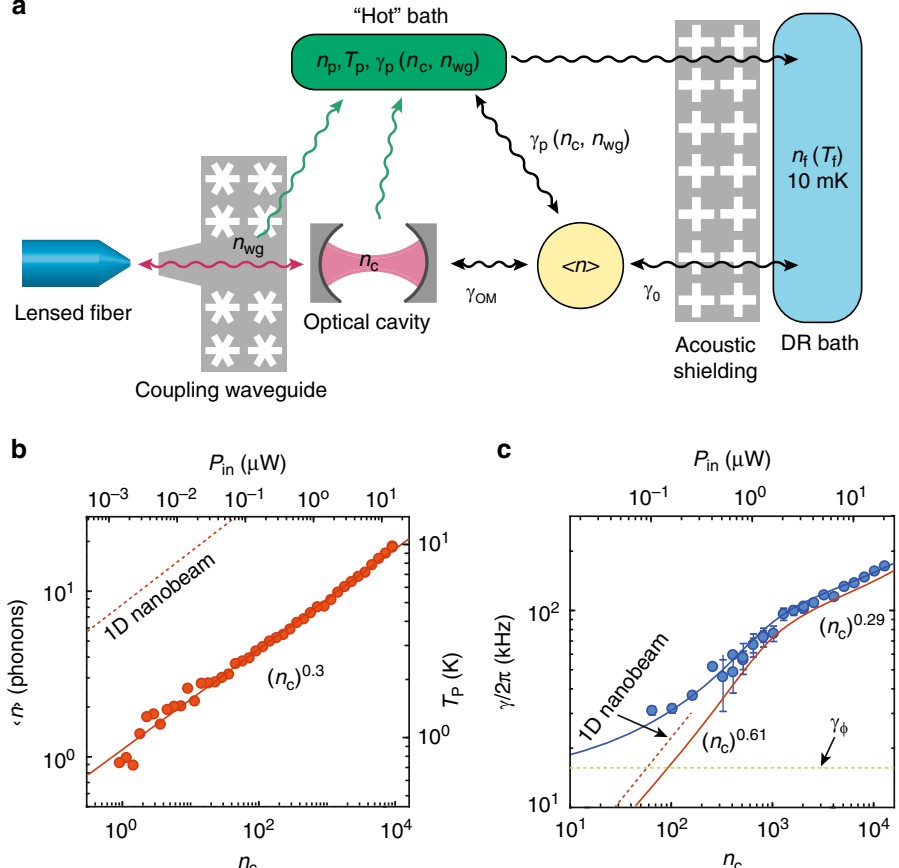

**Fig. 4 Optical-absorption-induced hot bath. a** Diagram illustrating the proposed model of heating of the mechanics due to optical absorption and the various baths coupled to the localized mechanical mode. **b** Plot of measured $n_p$ versus $n_c$. The solid line is a fit to the data giving $n_p = (1.1) \times n_c^{0.3}$. Dashed line indicates $n_p$ versus $n_c$ for a 1D nanobeam device measured in ref. [32] for comparison. **c** Plot of measured linewidth $\gamma$ (blue dots) versus $n_c$. The blue solid line is a power-law fit to the data, where in the low $n_c$ regime $\gamma/2\pi \approx \gamma_\phi/2\pi + \gamma_p/2\pi = 14.54$ kHz + (1.1 kHz) $\times n_c^{0.61}$ and in the high $n_c$ regime $\gamma/2\pi = 23.91$ kHz + (9.01 kHz) $\times n_c^{0.29}$. The red solid curve is the resulting fit for $\gamma_p$ by itself. For comparison, the dashed red curve is a plot of $\gamma_p$ versus $n_c$ for a 1D nanobeam device from ref. [32]. Error bars correspond to 90% confidence interval in fit to Lorentzian spectral linewidth (error bars smaller than the symbol are not shown). In **b**, **c** the quasi-2D OMC cavity measurements are for the same 8-shield device as in Fig. 3c.

laser pump field using a balanced heterodyne receiver (see Fig. 2d and "Methods"). In such a measurement, with resonant pump laser field ($\Delta = 0$), the acoustic mode linewidth is expected to contain contributions from the intrinsic energy decay rate $\gamma_0$, the optical absorption bath-induced damping $\gamma_p$, and any pure dephasing effects (frequency jitter) of the acoustic resonance $\gamma_\phi$. Referring to Fig. 4c, we see that the linewidth dependence on $n_c$ can be separated into three different regimes: (i) at the lowest powers ($n_c \lesssim 10$) the linewidth begins to saturate to a constant value (in our fit, this is given by $\gamma_\phi$ ($\gamma_0$ is entirely negligible on this scale), (ii) a low-power regime ($100 < n_c < n_{th} = 1000$) with a relatively strong dependence of linewidth on optical power, and (iii) a high-power regime ($n_c > n_{th} = 1000$) with a second, weaker dependence of linewidth on optical power, where $n_{th}$ is the threshold for $n_c$ used between (ii) and (iii) in the fitting (see Supplementary Note 7). Fitting the low-power regime with a power-law dependence on $n_c$, we find $\gamma/2\pi = \gamma_\phi/2\pi + (1.1 \text{ kHz}) \times n_c^{0.61}$, with $\gamma_\phi/2\pi = 14.54$ kHz. At these lower powers, we find a power-law scaling and overall magnitude of damping of the breathing mode of the quasi-2D OMC cavity which is close to that for the 1D nanobeam cavities of ref. [32] (see the dashed red curve in Fig. 4c). In the high-power regime, we find a fit given by $\gamma/2\pi = 23.91$ kHz + (9.01 kHz) $\times n_c^{0.29}$, with a power-law exponent that is approximately half that in the low-power regime. Determining the exact mechanism of the $\gamma_p$ slow

down versus $n_c$ (or indirectly $T_p$) in the high-power regime is outside the scope of this article; however, possibilities include a change in the phonon scattering rate with increasing phonon frequency in the nanostructured Si film[55,56] or a transition from Landau–Rumer scattering to Akhiezer-type damping as the effective bath temperature rises[57].

Before moving on to measurements of back-action cooling in the quasi-2D OMC cavity, we note one important distinction between the geometry of the optical coupling in the new 2D devices in comparison to previously studied 1D nanobeam devices. Whereas in the 1D nanobeam devices the coupling waveguide is evanescently coupled to the OMC cavity—and is thus not in direct mechanical contact with the nanobeam cavity—in the quasi-2D devices the optical coupling waveguide is physically connected to one end of the OMC cavity region (see Fig. 2a). Optical absorption in the coupling waveguide of the quasi-2D OMC devices may thus also lead to heating of the acoustic cavity mode. This effect is further corroborated by FEM simulations, detailed in Supplementary Note 8, that show that a weak cavity is formed between the end of the coupling waveguide and the quasi-2D OMC cavity. Optical absorption of the input power ($P_{in}$) in the coupling waveguide can be modeled as an effective waveguide photon number, $n_{wg}$, where $n_{wg} = \beta P_{in}$ for some fixed constant $\beta$ independent of cavity detuning. Assuming that the dependence of $n_p$ on $n_{wg}$ is the same as that for $n_c$, we can

write $n_p(n_c, P_{in}) = n_p(n_c + n_{wg})$. Similarly, $\gamma_p(n_c, P_{in}) = \gamma_p(n_c + n_{wg})$. In the measurements above with resonant pumping at $\Delta = 0$, very small input powers were required to build up large intra-cavity photon numbers, and as such $n_c \gg n_{wg}$. In what follows, where we perform back-action cooling with $\Delta = \omega_m \gg \kappa$, the input power required to yield a given $n_c$ is much larger and $n_{wg}$ cannot be ignored.

**Effective quantum cooperativity.** The ability of a cavity opto-mechanical systems to perform coherent quantum operations between the optical and mechanical degrees of freedom requires both large cooperativity $C \equiv \gamma_{OM}/\gamma_b$ and a mechanical mode thermal occupancy $\langle n \rangle < 1$ (the thermal noise in the optical mode is assumed negligible)[5,23]. Here $\gamma_b$ represents the total coupling rate of the mechanical system to its various thermal baths, which in the case of the Si OMC cavities, is given by $\gamma_b = \gamma_0 + \gamma_p$. The relevant figure of merit is then the effective quantum coopera-tivity $C_{eff} \equiv C/n_b^2$, where $n_b$ is the total effective bath occupancy defined by the relation $\gamma_b n_b \equiv \gamma_0(n_0 + 1) + \gamma_p(n_p + 1)$, where the "+1" terms correspond to spontaneous decay and $n_0 \lesssim 10^{-3}$ is the bath occupancy in the surrounding chip region of the OMC cavity (see Supplementary Note 9). As $\gamma_0 \ll \gamma_p$ for the optical powers used in this work and $n_0 \ll 1$, in what follows $\gamma_b n_b \approx \gamma_p(n_p + 1)$.

A measurement of the quantum cooperativity $C_{eff}$ can be made by observing the cooled mechanical occupancy under optical back-action cooling of the coupled mechanical mode,

$$\langle n \rangle = \frac{\gamma_p n_p + \gamma_0 n_0}{\gamma_0 + \gamma_{OM} + \gamma_p} = \frac{n_b - 1}{C + 1} \stackrel{n_b, C \gg 1}{\approx} \frac{1}{C_{eff}}, \qquad (1)$$

where we have implicitly assumed that the optical pump laser responsible for producing back-action damping $\gamma_{OM}$ has a zero effective noise occupancy. Back-action cooling is most efficient in the resolved sideband limit ($\kappa/2\omega_m < 1$) when the optical pump is applied on the red-detuned motional sideband of the cavity, $\Delta = \omega_m$. In Fig. 5a, we show the measured cooling curve of the quasi-2D snowflake cavity under continuous-wave optical pumping at $\Delta = \omega_m$. We infer the measured 10.2 GHz acoustic cavity mode occupancy (blue dots) from calibration of the photon counts of the anti-Stokes sideband of the reflected optical pump laser (see Fig. 2d and "Methods").

We can also predict the back-action cooling curve of the acoustic mode based on our independent measurements of $\gamma_{OM}$ (Fig. 3b), $\gamma_p$ (Fig. 4b), and $n_p$ (Fig. 4c) versus $n_c$. Using Eq. (1), we plot the theoretical acoustic mode cooling versus $n_c$ (i.e., $\beta = 0$) as a solid red curve in Fig. 5a. Not only does the $\beta = 0$ theoretical curve predict substantially more cooling of the acoustic mode than measured but also the shape of the measured and theoretical curves are quite different. As alluded to at the end of the previous section, one significant difference between the back-action cooling measurements and the measurements of $n_p$ and $\gamma_p$ is that the cooling measurements are performed at $\Delta = \omega_m$, requiring a 100-fold increase in the optical pump power to reach a given intra-cavity photon number $n_c$. For reference, we have plotted the corresponding input power $P_{in}$ on the top horizontal axes of Figs. 4b, c and 5a. Adding an additional waveguide photon number $n_{wg} = \beta P_{in}$ to the intra-cavity photon number $n_c$ in determining $n_p$ and $\gamma_p$, we find a modified cooling curve ($\beta = 15$ $\mu W^{-1}$; solid blue curve) that fits both the magnitude and shape of the measured cooling curve. In Fig. 5b, we plot the corresponding quantum cooperativity curve showing that $C_{eff}$ reaches above unity.

Although we have achieved $C_{eff} > 1$ under continuous-wave optical driving in the newly designed quasi-2D OMC cavities, looking forward, significant further increases can be achieved.

One clear method is to thermally decouple the input coupling waveguide from the OMC cavity in order to eliminate the parasitic heating from $n_{wg}$. This can be accomplished, for instance, by using evanescent side-coupling instead of butt-coupling of the coupling waveguide to the cavity. A second approach to dramatically improving $C_{eff}$ is through improve-ments in optical quality factor, where similar quasi-2D planar photonic crystal devices have already been demonstrated with optical $Q$-factors approaching $10^{7}$[58]. In Fig. 5c, we estimate achievable $\langle n \rangle$ and $C_{eff}$ as a function of $n_c$ and $Q_c$ assuming that $n_{wg}$ has been successfully eliminated. We find that, for a $Q$-factor of $Q_c = 3.90 \times 10^5$, equal to that of the zero-shield device of Fig. 3a, it should be possible to reach $\langle n \rangle \approx 0.1$ and $C_{eff} \approx 5$ for optical pump powers at the single photon level.

## Discussion

In conclusion, we have presented the design, fabrication, and characterization of a new quasi-2D OMC cavity with a breathing-like acoustic mode of 10 GHz frequency and large vacuum optomechanical coupling rate $g_0/2\pi \gtrsim 1$ MHz. By employing an engineered 2D phononic and photonic bandgap material in which to host the OMC cavity, the acoustic breathing-like mode of interest is well protected from its environment, while phonon modes above the acoustic bandgap serve as additional channels for removing heat from the cavity region. Through this dual role of the 2D bandgap structure, we demonstrate at millikelvin temperatures a localized acoustic cavity mode with intrinsic $Q$-factor of $1.2^{+0.12}_{-0.15} \times 10^9$ and a greatly increased ($\times 68$) thermal conductance between the cavity and the cold bath reservoir of the surrounding chip compared to previous 1D nanobeam OMC devices. These properties of the quasi-2D OMC cavity allow us to achieve a quantum cooperativity $C_{eff} > 1$ under continuous-wave optical pumping. They also point the way for significant further improvements in quantum cooperativity through modification in the optical coupling geometry.

This result ushers forth a variety of quantum optomechanical applications using chip-scale OMCs. In particular, in the case of hybrid superconducting and acoustic microwave quantum circuits[33,59–61], optomechanical devices that can operate in con-tinuous mode in the high $C_{eff}$ regime at millikelvin temperatures would enable bi-directional conversion of itinerant optical and microwave quantum signals (see Supplementary Note 10), forming the critical interface necessary to realize an optical quantum network[62] of superconducting quantum circuit nodes. These advances may also allow quantum optomechanical mea-surements to be performed at even lower temperatures than currently possible with OMC cavities, which, given their demonstrated ultralow rates of intrinsic acoustic dissipation[32], will allow for further studies of theories related to gravitationally induced decoherence[63,64] and nonlinearities and dephasing in mechanical systems[65].

## Methods

**Device fabrication.** The devices were fabricated from a SOI wafer (SEH, 220 nm Si device layer, 3 μm buried-oxide layer) using electron beam lithography followed by inductively coupled plasma reactive ion etching. The Si device layer is then masked by photoresist to define a "trench" region of the chip to be etched and cleared to which a lens fiber can access the chip coupling waveguides. In the unprotected trench region of the chip, the buried-oxide layer is removed with a highly aniso-tropic plasma etch, and the handle Si layer is removed to a depth of 120 μm using an isotropic plasma etch. The devices were then released in vapor hydrofluoric acid (HF) and cleaned in a piranha solution (3:1 $H_2SO_4$:$H_2O_2$) before a final diluted HF etch to remove any surface oxides.

**Device characterization.** Fabricated devices are characterized using a fiber-cou-pled, wavelength-tunable external cavity diode laser. The laser light is sent through a 50-MHz bandwidth-tunable fiber Fabry–Perot filter (Micron Optics FFP-TF2) to reject laser phase noise at the mechanical frequency. After this prefiltering, the light

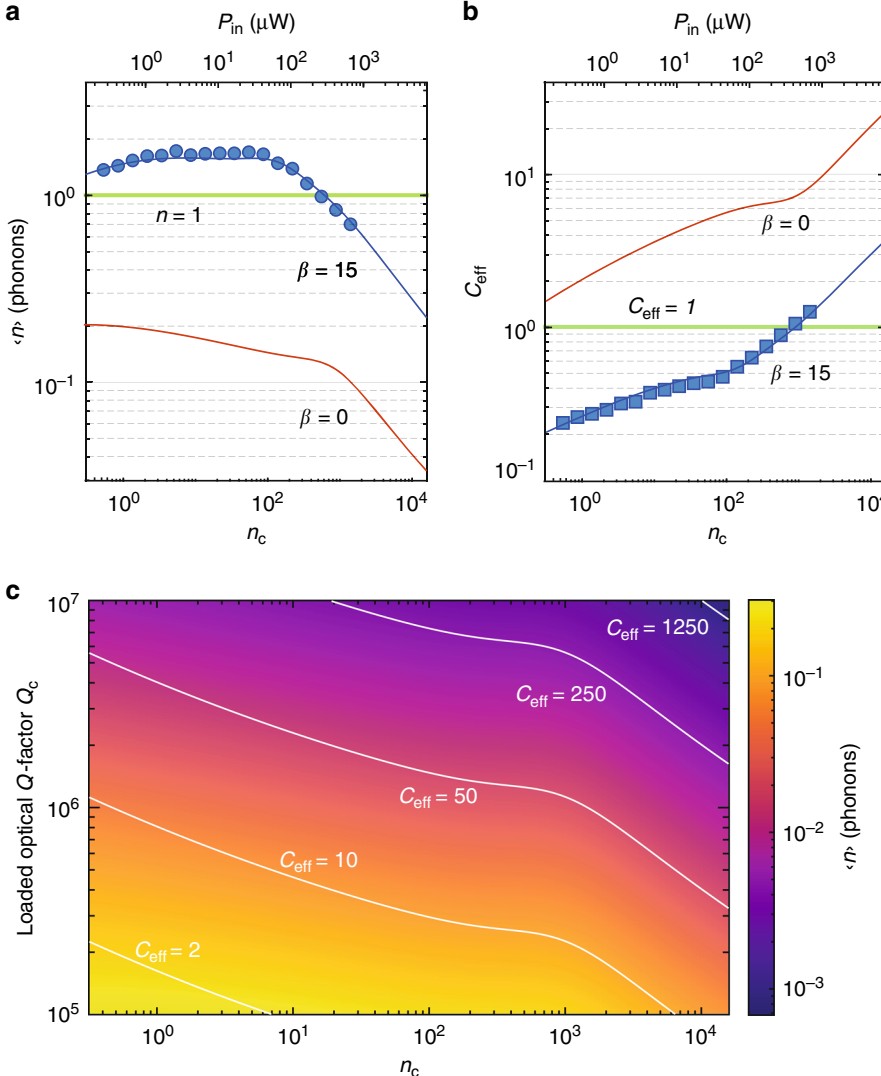

**Fig. 5 Phonon occupancy and effective quantum cooperativity. a** Plot of the measured occupancy $\langle n \rangle$ in the acoustic mode at $\omega_m/2\pi = 10.02$ GHz versus laser pump power in units of intra-cavity photons, $n_c$, and optical power coupled into coupling waveguide, $P_{in}$. Filled blue circles are measured data. **b** Plot of quantum cooperativity $C_{eff}$ versus $n_c$ and $P_{in}$. The $C_{eff}$ data points (filled blue squares) are inferred using Eq. (1) from the measured $\langle n \rangle$ in **a** along with the fit to the measured $n_p$ and $\gamma_p$ from Fig. 4. In both **a** and **b**, the solid curves are theoretical plots calculated using the power-law fits of $\gamma_p$ and $n_p$ considering optical heating from $n_c$ alone ($\beta = 0$, red solid line curve) and including direct heating from $P_{in}$ with $\beta = 15$ μW⁻¹ (blue solid curve). **c** Estimated back-action cooled phonon occupancy $\langle n \rangle$ for a quasi-2D OMC device with the same properties as measured in **a** save for a modified $Q$-factor and no waveguide heating ($\beta = 0$). White solid curves delineate contours of constant quantum cooperativity $C_{eff}$. In **a**, **b**, quasi-2D OMC cavity measurements are for the same device as in Figs. 3c and 4.

path can be switched by 2 × 2 optical switches between two paths: (i) a "balanced heterodyne spectroscopy" path with a high-speed photodetector for performing balanced heterodyne detection of the acoustic mode, and (ii) a "photon counting path" with a single photon detector for performing either pulsed or continuous-wave phonon counting of the acoustic mode.

For the balanced heterodyne spectroscopy path, a 90:10 beam splitter divides the laser source into local oscillator (LO; 90%) and signal (10%) beams. The LO is modulated by an electro-optic modulator (EOM) to generate a sideband at $\delta/2\pi = 50$ MHz from the mechanical frequency. One of the modulated sidebands of LO is selected using a high-finesse tunable Fabry–Perot filter before recombining it with the signal for detection. The signal path is sent through a variable optical attenuator and then to an optical circulator, which directs the signal to the device under test in the DR. A lensed optical fiber in the DR is used to couple light into and out of the devices. The reflected signal beam is recombined with the LO using a variable optical coupler, the outputs of which are sent to a balanced photodetector.

For the photon counting path, the laser is directed via 2 × 2 mechanical optical switches into a "high-extinction" branch consisting of an acousto-optic modulator (AOM; 20 ns rise and fall time, 50 dB on–off ratio) and two Agiltron NS switches (Ag.; 100 ns rise time, 30 μs fall time, total of 36 dB on–off ratio), which are driven by a digital delay generator to generate high-extinction-ratio optical pulses. The digital delay generator is used to synchronize the switching of the AOM and

Agiltron switches with the time-correlated single-photon-counting module, which is connected to a single-photon detector (SPD). The SPDs used in this work are amorphous WSi-based superconducting nanowire single-photon detectors. The tunable fiber Fabry–Perot filters used for both prefiltering the pump and filtering the motionally generated sidebands have a bandwidth of 50 MHz, a free-spectral range of 20 GHz, and a tuning voltage of ≤18 V per free-spectral range. Each of the filters provide 40 dB extinction at 10 GHz offset from the transmission peak.

**Calibration of $n_c$ and $g_0$.** The photon number $n_c$ at a given power and detuning depends on the single-pass fiber-to-waveguide coupling efficiency $\eta_{cpl}$ and cavity extrinsic coupling efficiency $\eta_\kappa = \kappa_e/\kappa$. The fiber-to-waveguide coupling efficiency $\eta_{cpl}$ was determined by measuring the reflection-level far off-resonance from the optical cavity using a calibrated optical power meter and was found to be $\eta_{cpl} = 0.59$ for the zero-shield device and $\eta_{cpl} = 0.6$ for the 8-shield device of this work. The cavity extrinsic coupling efficiency $\eta_\kappa$ was measured by placing the frequency of the pump laser far off-resonance and using a vector network analyzer (VNA) to drive an EOM to sweep an optical sideband through the cavity frequency. The optical response is measured on a high-speed photodiode that is connected to the VNA signal port. The amplitude and phase response of the cavity are obtained and fit to determine $\eta_\kappa$ and $\kappa$. With these two parameters, we determine $n_c$ for a given

input power,

$$n_c = \frac{P_{in}}{\hbar \omega_p} \frac{\kappa_e}{\Delta^2 + (\kappa/2)^2}, \qquad (2)$$

where $\omega_p$ is the pump laser frequency. To extract the vacuum optomechanical coupling rate $g_0$, we measure the acoustic mode linewidth versus optical power for $\Delta = \omega_m$. The slope of the linewidth versus the calibrated $n_c$ yields $4g_0^2/\kappa$, from which we determine $g_0$. At millikelvin temperatures, we use the measured per-phonon scattering rate, $\Gamma_{SB,0}$, in a similar fashion to determine $g_0$ (see below).

**Ringdown measurements.** The intrinsic mechanical $Q$-factor of the quasi-2D OMC devices is measured using a ringdown technique. The ringdown measurement we employ uses pulsed optical excitation and photon counting techniques as presented previously in refs. [32,53]. In the simplest version, the measurement relies on generating a train of optical laser pulses detuned to the lower-frequency motional sideband of the optical cavity ($\Delta = \omega_m$), with excitation and read-out performed by the same pulse. Excitation of the acoustic mode into a low-occupancy thermal state (a few phonons) is provided by the optical absorption heating during the pulse. Read-out of the acoustic mode occupancy is performed by photon counting of the anti-Stokes sideband photons of the pump as described below in the discussion of the back-action cooling measurements.

For ringdown measurements in this work, we use a train of 10-μs-long high-extinction optical laser pulses with the laser frequency tuned to the red-motional sideband of the OMC cavity resonance ($\Delta = \omega_m$). The pulses are generated as described above in the description of the photon counting path of the measurement set-up, with an on–off extinction exceeding 80 dB and rise and fall times of approximately 20 ns. We used a peak pulse power corresponding to $n_c = 60$. For each $\tau_{off}$, we average over many optical pulses the ratio of the inferred mode occupancy within the first 25 ns time bin of the optical pulse ($n_i$) to that of the inferred occupancy in the last 25 ns time bin of the optical pulse ($n_f$). The inter-pulse delay $\tau_{off}$ is varied to map out the decay of the acoustic mode energy in between optical pulses.

**Measurement of hot bath occupancy $n_p$.** Measurements of the occupancy of the hot bath are performed using the photon counting path of the measurement set-up, with the pump laser operated in continuous wave (i.e., no modulation) and tuned to cavity resonance ($\Delta = 0$). The reflected pump laser from a device under test is filtered at an offset equal to the mechanical mode frequency to separate the motionally generated anti-Stokes sideband from the pump. This filtered signal is then sent to the SPD for detection. The measured photon count rate for this pump laser detuning is given by $\Gamma(\Delta = 0) = \eta(\kappa/2\omega_m)^2 \gamma_{OM} \langle n \rangle$, where $\eta$ is the total optical detection efficiency of sideband photons, $\kappa$ is the total optical cavity mode decay rate, and $\gamma_{OM} = 4g_0^2 n_c/\kappa$ is the parametric optomechanical coupling rate. The observed count rate—in conjunction with calibration of $\eta$ and independent measurement of $\kappa$, $\omega_m$, and $\gamma_{OM}$—is then used to extract the mode occupancy of the coupled breathing-like mode, $\langle n \rangle$. Owing to the lack of optomechanical back-action damping for $\Delta = 0$ laser detuning, this measured occupancy is a close approximation to the hot bath occupancy $n_p$ at power levels where the effective hot bath temperature $T_p$ is much greater than the base temperature to which the mode thermalizes with no applied laser, $T_0 \approx 63$ mK (see Supplementary Note 9).

**Back-action cooling measurements.** Back-action cooling measurements are also performed using the photon counting path of the measurement set-up but with the optical pump laser detuned to the red-motional sideband of the optical cavity resonance ($\Delta = \omega_m$). An accurate measure of the acoustic phonon mode occupancy for the back-action cooling measurements can be determined by using vacuum noise as a reference meter. The measured photon count rate for a red- or blue-detuned pump laser is given by [53]:

$$\Gamma(\Delta = \pm\omega_m) = \Gamma_{DCR} + \Gamma_{pump} + \Gamma_{SB,0}\left(\langle n \rangle + \frac{1}{2}(1 \mp 1)\right), \qquad (3)$$

where $\Gamma_{DCR}$ is the dark count rate of the SPD, $\Gamma_{pump}$ is the count rate due to any bleed-through from the sideband filtering of the pump laser, and $\Gamma_{SB,0} = \eta_{det}\eta_{cpl}\eta_\kappa \gamma_{OM}$ is the detected photon scattering rate per phonon on the SPD. Here $\eta_{det}$ is the measured overall detection efficiency of the set-up, including insertion losses in the fibers inside and outside of the DR, fiber unions, fiber circulator, and the detection efficiency of the SPD ($\eta_{SPD}$). We then calibrate $\Gamma_{SB,0}$ using a pulsed blue-detuned laser pump ($\Delta = -\omega_m$). The inter-pulse delay ($\tau_{off}$) of the blue-detuned pulses was selected to be much longer than the intrinsic damping time of the acoustic mode such that at the beginning of the pulse $\langle n \rangle \approx n_0 \ll 1$ and the sideband photon count rate is set by spontaneous vacuum scattering of the pump, $\Gamma \approx \Gamma_{SB,0}$ ($\Gamma_{DCR} + \Gamma_{pump}$ was confirmed to be much smaller than $\Gamma_{SB,0}$). Normalizing the measured count rate by $\Gamma_{SB,0}$ in the back-action cooling measurements yields the calibrated phonon occupancy of the acoustic mode. Note that we also used this technique to calibrate the ringdown measurement of Fig. 3c in units of phonon number.

## Data availability

The data that support the findings of this study are available from the corresponding author (O.P.) upon reasonable request.

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

## Acknowledgements

The authors thank A. Sipahigil for valuable discussions. This work was supported by the AFOSR-MURI Quantum Photonic Matter, the ARO-MURI Quantum Opto-Mechanics with Atoms and Nanostructured Diamond (grant N00014-15-1-2761), the Institute for Quantum Information and Matter, an NSF Physics Frontiers Center with support of the Gordon and Betty Moore Foundation, and the Kavli Nanoscience Institute at Caltech. H.R. is supported by the National Science Scholarship from A*STAR, Singapore.

## Author contributions

H.R., G.S.M., and O.P. came up with the concept and planned the experiment. H.R., J.L., and H.P. performed the device design and fabrication. H.R., G.S.M., and M.M. performed the measurements. H.R., M.H.M., and O.P. analyzed the data. All authors contributed to the writing of the manuscript.

## Competing interests

The authors declare no competing interests.
