## [Peer Review File · Nature Communications]

Reviewers' comments:

Reviewer #1 (Remarks to the Author):

This paper provides a clear description of the design, fabrication and measurement of two-dimensional optomechanical crystals that are cooled close to their mechanical quantum ground state. Measurements are reported for the optical and mechanical loss, mechanical heating due to the optical probe signal, and estimates made for the quantum cooperativity for optomechanical state transfer.

This paper reports an astounding mechanical quality factor in excess of one billion at 10 GHz, reproducing, and closely parallel to, an earlier manuscript from the same group (ref 32), which dealt with low temperature behavior in one-dimensional optomechanical crystals. This earlier work, with even higher reported mechanical Q's at similar frequencies, apparently has not yet been published, which is unfortunate, as this earlier paper represents a signature result with a detailed analysis very similar to this work. The complicated thermal modeling and behavior reported here is quite similar to that earlier work.

In the submitted paper here a similar level of mechanical quality factor is achieved at the lowest optical excitation powers as in Ref 32, with this measurement similarly confounded by heating due to the laser probe. However in this two-dimensional design the heating is reduced sufficiently that more promising performance is predicted for somewhat more optimized devices. The paper is mostly very clear and shows a number of new results; the supplementary also provides a wealth of experimental detail that is quite welcome. I would support publication in Nature Communications with some mandatory changes as below.

Please provide an explicit expression for the cooperativity C when it is introduced on p 1

Please provide an expression and reference for the extraction of the optomechanical coupling rate g from the linewidth dependence on cavity photon number.

Many parameters are calculated from the data without any consideration for the uncertainties in the arrived-at results; just as an example, the mechanical decay rates given on p 6 are given with 4 significant figures; I find it hard to believe, given the data, that these can be determined to 1 part in 10^4 . Please provide uncertainties for all such derived quantities.

Reviewer #2 (Remarks to the Author):

The authors demonstrate a two-dimensional (2D) Si optomechanical crystal (OMC) cavity with a high quantum cooperativity at millikelvin temperatures. The structure is well band-engineered in both photonic and phononic systems and achieves high optical and mechanical quality factors. This group investigated similar topic in a 1D nanobeam OMC structure before. In this study, design and fabrication of a 2D OMC cavity with a high acoustic cavity Q and much better thermal conductance with the cold bath reservoir result in a higher quantum cooperativity, which is the most relevant figure-of-merit for quantum optomechanical applications. The progress from the 1D to 2D OMC cavity is clear as stated in the last paragraph in p. 1, and this work achieves an effective quantum cooperativity of unity. The level of experiments and analyses are high, and the discussion is reasonable and convincing. Therefore, I recommend publication after minor revision.

1. Regarding the quantum cooperativity, the relevant threshold for coherent photon-phonon interaction is unity. How high quantum cooperativity is necessary for realistic use in applications? Please pick up one example of application and discuss how far/close current technology level is.

2. In Fig. 4b, n_p is proportional to $n_c^{0.3}$. What determines this power-law of 0.3? Is it reasonable

that 1D and 2D have the same power-law?

3. (minor) In Fig. 1c, the broken lines are too thin and could not find them when I printed. I recommend making it thicker.

Reviewer #3 (Remarks to the Author):

I have read the manuscript « Two-Dimensional Optomechanical Crystal Cavity with High Quantum Cooperativity” from Prof Painter and co-workers. Briefly, the manuscript describes a new type of two-dimensional optomechanical crystal cavity engineered for improving the low temperature thermalization properties and thereby maximize the optomechanical cooperativity (which represents the weight of quantum fluctuations respective to that of classical ones). This work reports on the background, design and fabrication of the device, as well as on the optomechanical and thermal experimental characterizations and calibration. The context and hypothesis of the work are remarkably clear, as well as the adopted scientific methodology. The quality of the experimental results and their agreement with theoretical modelling are high. This manuscript represents an important piece of work to the field and beyond, thanks to the general problematic being addressed (heat bath engineering applied to the design of quantum coherent devices) and its very accessible presentation. In light of the above remarks, I strongly recommend publication of this major piece of work in Nature Communications.

The manuscript is organized into 6 parts: The authors thoroughly introduce the context of their work in the first part. Details on the design methodology and fabrication are given in the second part, along with the presentation of the experimental setup. The optomechanical characterization and calibration is provided in the third part. The fourth part presents the measurement and analysis of the thermal properties of the optomechanical system under continuous optical driving. The fifth part reports the measurement of the phonon occupation and corresponding effective quantum cooperativity under continuous optical drive. The authors conclude in the last, discussion part.

Introduction

In the first, introductory part, the authors set their work into the general context of (quantum) optomechanical systems. They clearly describe the importance of developing high cooperativity optomechanical systems operating at ultra-low temperatures, notably in the perspective of building quantum hybrid interfaces between microwave frequency logic circuits and optical quantum communication channels. The intrinsic problematic of 1-dimension optomechanical crystal cavity systems, showing very high optomechanical coupling rates but reduced thermal conductivity (and which therefore greatly suffer from the absorption-induced heating) is very well introduced.

Comment: I have no specific comment on this very well written part.

Design/fabrication

In the second part, the authors present the concept of their new device aiming at preserving a very high optomechanical coupling while simultaneously increasing the thermal conductivity, enabling to load significantly more intra-cavity photons, and therefore resulting in a much increased effective optomechanical cooperativity. Their strategy relies on making use of the frequency-dependent density of phonon states within a 2D phononic bandgap structure consisting of a 2D optomechanical crystal cavity. The fabrication steps, the analysis of the experimentally measured geometrical properties and the simulated optomechanical parameters are clearly presented in that section, confirming the relevance of the proposed approach as far as the optomechanical properties are concerned.

Comment: I have a minor comment regarding Fig 2.c: I would advise the authors to mind the use of colours (and to maybe compliment it with different point styles) for the colours-blind readership.

Optomechanical characterization

In this third part, the authors present the optomechanical characterization of the newly fabricated devices. Optomechanical characterization is reported both at room and cryogenic temperature,

including the optical Q-factor, mechanical resonance frequency, optomechanical coupling rate and mechanical Q-factor. In particular, the authors pay a great deal of attention for avoiding dynamical backaction effects by performing ringdown measurements, which besides showing decreased sensitivity towards dephasing, enables to be "in the dark", thereby suppressing the contribution of any delayed dynamical backaction. The authors notably report a massive mechanical Q-factor exceeding 1 billion, with a mechanical resonance frequency in the 10 GHz range.

Comment: I have a comment on this part: The authors report measurements relying on non-linear optomechanical amplification of a resonant phase modulation, which is sometimes also referred to as "OMIT" measurement. I believe such experiment to be nontrivial to a broad readership and would suggest a more pedagogic presentation, besides the (rightfully) given references. Along these lines, I would recommend a description of the solid line fits featured on Fig. 3(b).

Thermal measurements

In this fourth part, the authors present a study of the thermal properties (both effective damping rate and phonon occupation) of the OMC cavity as a function of the input optical power. The authors essentially identify two regimes for the sensitivity of the mechanical properties as a function of the intracavity photon number. Importantly, they establish a connection between the effective temperature (number of phonons) and the mechanical damping rate, thereby describing the effect of the absorption as that of an effective "hot bath".

Comments: I have a few comments on this part.

a) I find the last sentence of the second paragraph (right column) on page 5 somehow too long and not easy to understand. I would recommend the authors to try to simplify this sentence. On the power low: could the authors maybe comment on a more fundamental solid-state physics point of view maybe?

b) 3rd paragraph right column on page 5: the authors refer to Fig. 3(b) instead of Fig. 4(b).

c) page 6 right column: The authors state "at the lowest power (...) the linewidth saturates to a constant value; this is not entirely clear to me that the data confirm this.

d) On γ_{ϕ} : how do the authors prove this to be dephasing (besides being much larger than the "zero power" ring down value)? Did the authors maybe perform ringdown measurements (e.g. by means of a pump-probe configuration)? Could the authors better explain why the two regimes of temperature are fitted using different models? Where do the authors set the "cut off" between these two regimes?

e) End of 2nd paragraph right column page 6: The last sentence sounds somehow cryptic.

Measurement of the optomechanical cooperativity

In the fifth part, the authors report on optomechanical thermometry measurements, and corresponding optomechanical cooperativity. Their notably use their analysis to attribute the (significant) excess of thermal occupation to the phonon population within the coupling waveguide, which contaminates that of the cavity under the effect of optical absorption. The (somehow empiric) model of the authors shows very satisfying agreement, which certainly paves the way not only to further technological improvement but for deeper understanding and assessment of the spatial location of decoherence in ultra-sensitive optomechanical systems.

Comments: I have a comment/question on this part: Page 7 left column 3rd paragraph : did the authors tried to optimize the sideband ratio to see if this could be beneficial to the cooperativity?

Discussion

In the last part, the authors briefly discuss their result and put them into perspective with future possible research pathways that could benefit from them. This part is convincing and very well written.

Comment: I have one minor comment: An 68 fold increase of the thermal conductivity is mentioned, which disagrees with the number (42) given in the main text body.

Author Response:

The authors thank all the reviewers for their careful and detailed review of our manuscript. Please find below the authors' responses (**marked in red**) to each of the reviewers' points (**in black**).

Reviewer #1 (Remarks to the Author):

This paper provides a clear description of the design, fabrication and measurement of two-dimensional optomechanical crystals that are cooled close to their mechanical quantum ground state. Measurements are reported for the optical and mechanical loss, mechanical heating due to the optical probe signal, and estimates made for the quantum cooperativity for optomechanical state transfer.

This paper reports an astounding mechanical quality factor in excess of one billion at 10 GHz, reproducing, and closely parallel to, an earlier manuscript from the same group (ref 32), which dealt with low temperature behavior in one-dimensional optomechanical crystals. This earlier work, with even higher reported mechanical Q's at similar frequencies, apparently has not yet been published, which is unfortunate, as this earlier paper represents a signature result with a detailed analysis very similar to this work. The complicated thermal modeling and behavior reported here is quite similar to that earlier work.

In the submitted paper here a similar level of mechanical quality factor is achieved at the lowest optical excitation powers as in Ref 32, with this measurement similarly confounded by heating due to the laser probe. However in this two-dimensional design the heating is reduced sufficiently that more promising performance is predicted for somewhat more optimized devices. The paper is mostly very clear and shows a number of new results; the supplementary also provides a wealth of experimental detail that is quite welcome. I would support publication in Nature Communications with some mandatory changes as below.

Please provide an explicit expression for the cooperativity C when it is introduced on p 1

[Authors' Response]: Thanks for pointing this out. We have added an expression for the cooperativity C when it is introduced in the fourth paragraph on p1.

Please provide an expression and reference for the extraction of the optomechanical coupling rate g from the linewidth dependence on cavity photon number.

[Authors' Response]: Thanks for the comment. An expression and reference for the extraction of the optomechanical coupling rate g from the linewidth dependence on cavity photon number (Fig. 3(b) solid line fit) have been added in the manuscript (first paragraph of page 4).

Many parameters are calculated from the data without any consideration for the uncertainties in the arrived-at results; just as an example, the mechanical decay rates given on p 6 are given with 4 significant figures; I find it hard to believe, given the data, that these can be determined to 1 part in 10^4 . Please provide uncertainties for all such derived quantities.

[Authors' Response]: Yes, we agree with the reviewer. We have added uncertainties to key device parameters with 90% confidence interval. For example, mechanical decay rate is reported as $8.28(+1.25,-0.43)$ ~Hz now.

Reviewer #2 (Remarks to the Author):

The authors demonstrate a two-dimensional (2D) Si optomechanical crystal (OMC) cavity with a high quantum cooperativity at millikelvin temperatures. The structure is well band-engineered in both photonic and phononic systems and achieves high optical and mechanical quality factors. This group investigated similar topic in a 1D nanobeam OMC structure before. In this study, design and fabrication of a 2D OMC cavity with a high acoustic cavity Q and much better thermal conductance with the cold bath reservoir result in a higher quantum cooperativity, which is the most relevant figure-of-merit for quantum optomechanical applications. The progress from the 1D to 2D OMC cavity is clear as stated in the last paragraph in p. 1, and this work achieves an effective quantum cooperativity of unity. The level of experiments and analyses are high, and the discussion is reasonable and convincing. Therefore, I recommend publication after minor revision.

1. Regarding the quantum cooperativity, the relevant threshold for coherent photon-phonon interaction is unity. How high quantum cooperativity is necessary for realistic use in applications? Please pick up one example of application and discuss how far/close current technology level is.

[Authors' Response]: One example application for cavity-optomechanical systems is phonon-mediated quantum state transduction between microwave and optical photons. We have added a section in the Supplementary Information discussing the bi-directional transduction efficiency and signal-to-noise ratio. This example highlights the fact that already at a $C_{eff} > 1$, one to in principle can realize single photon conversion with an $SNR > 1$.

2. In Fig. 4b, n_p is proportional to $n_c^{0.3}$. What determines this power-law of 0.3? Is it reasonable that 1D and 2D have the same power-law?

[Authors' Response]: This is a good point. The power law exponent α is equal to the effective number of spatial dimensions d of the material/structure under consideration. Effectively, the hot phonon bath radiates energy as a black body, with radiated power scaling as $T_p^{(\alpha+1)}$ via Planck's law, where T_p is the effective temperature of the "hot bath". In the case of a structure with 2-dimensional phonon density of states, $\alpha = d = 2$ and the hot phonon bath occupancy/temperature scales as $n_p \sim T_p \sim P_{in}^{1/3} \sim n_c^{1/3}$. This approximate scaling is expected to be valid so long as phonons in the hot phonon bath approximately thermalize with each other upon creation from optical absorption events, and then radiate freely (ballistically) into the effective zero temperature substrate.

The density of states for 1D and 2D OMC cavities are both determined by the nano-structure of the devices and the frequency of the phonons involved in heat transport. Due to the geometric aspect ratio of the thin-film (220nm membrane), the local density of phonon states becomes restricted at lower frequency, decreasing the rates of phonon-phonon scattering at low frequency relative to those of a bulk crystal with a 3D Debye density of states. A phonon bottleneck occurs as the density of states passes from 3D (continuum) to 2D due to this reduction of the phonon-phonon scattering. The phonons of the bath tend to pile up at these frequencies, and as such most of the phonons that contribute to thermal conduction will be around 20GHz corresponding to an acoustic wavelength of the thickness of the Si device layer (200nm). This Si device layer thickness is the smallest dimension of both the 1D and 2D OMCs, with the lateral dimension of the 1D OMC still much larger (x5) than this wavelength. As a result, both the 1D and 2D OMCs both have a hot phonon bath with approximately 2D density of states. Numerical simulations of the acoustic modes of the 1D nanobeam OMC confirm that above the OMC bandgap frequencies the density of phonon states is approximately that of a 2D plate. This detailed analysis is presented in

another of our papers which we reference, arXiv:1901.04129 (2019), Appendix D.1 and Appendix H.

3. (minor) In Fig. 1c, the broken lines are too thin and could not find them when I printed. I recommend making it thicker.

[Authors' Response]: Thanks for pointing this out. We have improved Fig. 1c (and also Fig. 1b) with thicker lines and higher contrast colors.

Reviewer #3 (Remarks to the Author):

I have read the manuscript « Two-Dimensional Optomechanical Crystal Cavity with High Quantum Cooperativity” from Prof Painter and co-workers. Briefly, the manuscript describes a new type of two-dimensional optomechanical crystal cavity engineered for improving the low temperature thermalization properties and thereby maximize the optomechanical cooperativity (which represents the weight of quantum fluctuations respective to that of classical ones). This work reports on the background, design and fabrication of the device, as well as on the optomechanical and thermal experimental characterizations and calibration. The context and hypothesis of the work are remarkably clear, as well as the adopted scientific methodology. The quality of the experimental results and their agreement with theoretical modelling are high. This manuscript represents an important piece of work to the field and beyond, thanks to the general problematic being addressed (heat bath engineering applied to the design of quantum coherent devices) and its very accessible presentation. In light of the above remarks, I strongly recommend publication of this major piece of work in Nature Communications.

The manuscript is organized into 6 parts: The authors thoroughly introduce the context of their work in the first part. Details on the design methodology and fabrication are given in the second part, along with the presentation of the experimental setup. The optomechanical characterization and calibration is provided in the third part. The fourth part presents the measurement and analysis of the thermal properties of the optomechanical system under continuous optical driving. The fifth part reports the measurement of the phonon occupation and corresponding effective quantum cooperativity under continuous optical drive. The authors conclude in the last, discussion part.

Introduction

In the first, introductory part, the authors set their work into the general context of (quantum) optomechanical systems. They clearly describe the importance of developing

high cooperativity optomechanical systems operating at ultra-low temperatures, notably in the perspective of building quantum hybrid interfaces between microwave frequency logic circuits and optical quantum communication channels. The intrinsic problematic of 1-dimension optomechanical crystal cavity systems, showing very high optomechanical coupling rates but reduced thermal conductivity (and which therefore greatly suffer from the absorption-induced heating) is very well introduced.

Comment: I have no specific comment on this very well written part.

Design/fabrication

In the second part, the authors present the concept of their new device aiming at preserving a very high optomechanical coupling while simultaneously increasing the thermal conductivity, enabling to load significantly more intra-cavity photons, and therefore resulting in a much increased effective optomechanical cooperativity. Their strategy relies on making use of the frequency-dependent density of phonon states within a 2D phononic bandgap structure consisting of a 2D optomechanical crystal cavity. The fabrication steps, the analysis of the experimentally measured geometrical properties and the simulated optomechanical parameters are clearly presented in that section, confirming the relevance of the proposed approach as far as the optomechanical properties are concerned.

Comment: I have a minor comment regarding Fig 2.c: I would advise the authors to mind the use of colours (and to maybe compliment it with different point styles) for the colours-blind readership.

[Authors' Response]: Thanks for pointing this out. We have improved Fig. 2c by tuning the colors used in this figure to a more color-blind-friendly, as well as using different symbols for each curve. We adjusted the description in the caption.

Optomechanical characterization

In this third part, the authors present the optomechanical characterization of the newly fabricated devices. Optomechanical characterization is reported both at room and cryogenic temperature, including the optical Q-factor, mechanical resonance frequency, optomechanical coupling rate and mechanical Q-factor. In particular, the authors pay a great deal of attention for avoiding dynamical backaction effects by performing ringdown measurements, which besides showing decreased sensitivity towards dephasing, enables to be “in the dark”, thereby suppressing the contribution of any delayed dynamical backaction. The authors notably report a massive mechanical Q-factor exceeding 1 billion, with a mechanical resonance frequency in the 10 GHz range.

Comment: I have a comment on this part: The authors report measurements relying on non-linear optomechanical amplification of a resonant phase modulation, which is sometimes also referred to as “OMIT” measurement. I believe such experiment to be nontrivial to a broad readership and would suggest a more pedagogic presentation, besides the (rightfully) given references. Along these lines, I would recommend a description of the solid line fits featured on Fig. 3(b).

[Authors' Response]: Thank you for the comment. We have added a section in the Supplementary Information explaining the concept of “OMIT”. An expression and description for the extraction of the optomechanical coupling rate g from the linewidth dependence on cavity photon number (Fig. 3(b) solid line fits) have been added in the manuscript.

Thermal measurements

In this fourth part, the authors present a study of the thermal properties (both effective damping rate and phonon occupation) of the OMC cavity as a function of the input optical power. The authors essentially identify two regimes for the sensitivity of the mechanical properties as a function of the intracavity photon number. Importantly, they establish a connection between the effective temperature (number of phonons) and the mechanical damping rate, thereby describing the effect of the absorption as that of an effective “hot bath”.

Comments: I have a few comments on this part.

a) I find the last sentence of the second paragraph (right column) on page 5 somehow too long and not easy to understand. I would recommend the authors to try to simplify this sentence. On the power low: could the authors maybe comment on a more fundamental solid-state physics point of view maybe?

[Authors' Response]: Point taken. We have worked to rephrase and simplify this sentence. The power law exponent is dependent on the effective number of spatial dimensions d of the material/structure under consideration. The density of states for the hot phonon bath in both 1D and 2D OMC cavities are both determined by the smallest dimension of each structure, thickness of the Si device layer. More details are provided in response to reviewer #1, comment 2.

b) 3rd paragraph right column on page 5: the authors refer to Fig. 3(b) instead of Fig. 4(b).

[Authors' Response]: Thank you for pointing this out. This was an oversight, and we have corrected it from 3(b) to 4(b).

c) page 6 right column: The authors state “at the lowest power (...) the linewidth saturates to a constant value; this is not entirely clear to me that the data confirm this.

[Authors' Response]: We agree with the reviewer, the wording was not accurate. The numbers measured directly do not get into the lowest power regime where we see complete saturation, rather the dependence on n_c **begins** to saturate. We have made this more explicit in the revised text to avoid confusion. However, we still apply in our model $\gamma(n_c) = \gamma_{\phi} + \gamma_p(n_c)$, and the best fit of this model gives a residual linewidth of γ_{ϕ} as indicated.

We should note that this pure dephasing term is expected, as for previous experiments with a similar OMC cavity (arXiv:1901.04129 (2019)), direct measurements of mechanical frequency jittering were performed, yielding γ_{ϕ} of a similar magnitude.

d) On γ_{ϕ} : how do the authors prove this to be dephasing (besides being much larger than the “zero power” ring down value)? Did the authors maybe perform ringdown measurements (e.g. by means of a pump-probe configuration)? Could the authors better explain why the two regimes of temperature are fitted using different models? Where do the authors set the “cut off” between these two regimes?

[Authors' Response]: In these devices we did not prove that the residual linewidth at low optical pumping was indeed due to dephasing (frequency jitter). However, as noted above, for very similar 1D OMC devices we did do rapid spectroscopic measurements using a pump-probe technique, and we were able to measure the frequency jitter of the line directly (arXiv:1901.04129 (2019)).

Regarding how we did the fitting across the different regimes, we have added a section in the Supplementary Information describing these details and explaining how the cut-off is implemented in our fit.

e) End of 2nd paragraph right column page 6: The last sentence sounds somehow cryptic.

[Authors' Response]: It is a rather simple (obvious) point, maybe that is why it seemed confusing. The difference is in the laser detuning, in one case on-resonance (both measurements) and in the other case far off resonance (back-action cooling). Because of this the input power required to put the same number of photons into the cavity is VERY different. Since n_{wg} scales with input power and not detuning, this means for the back-action cooling with large detuning and large input power, n_{wg} is much larger and cannot be ignored. We think the statement is clear as is in the manuscript.

Measurement of the optomechanical cooperativity

In the fifth part, the authors report on optomechanical thermometry measurements, and corresponding optomechanical cooperativity. They notably use their analysis to attribute the (significant) excess of thermal occupation to the phonon population within the coupling waveguide, which contaminates that of the cavity under the effect of optical absorption. The (somehow empiric) model of the authors shows very satisfying agreement, which certainly paves the way not only to further technological improvement but for deeper understanding and assessment of the spatial location of decoherence in ultra-sensitive optomechanical systems.

Comments: I have a comment/question on this part: Page 7 left column 3rd paragraph : did the authors tried to optimize the sideband ratio to see if this could be beneficial to the cooperativity?

[Authors' Response]: Yes, the cooperativity can be affected by the sideband ratio. There are two ways in which the cooperativity could increase with changes in the sideband ratio: (i) we could reduce the mechanical frequency which will reduce the sideband ratio, but also allow one to use lower input power to obtain the same internal cavity photon number (all things being equal, this would reduce the heating effects due to n_{wg}), or (ii) we could decrease the intrinsic damping of the cavity (κ_i) which would increase the sideband ratio, but also increase the back-action per cavity photon and thus the cooperativity. As you can see, it is not so much sideband ratio that directly matters for cooperativity, but rather mechanical frequency (fixed for our design), and the intrinsic cavity damping or intrinsic Q. We do mention improvements of intrinsic Q as being a key way forward to improving C_{eff} .

Discussion

In the last part, the authors briefly discuss their result and put them into perspective with future possible research pathways that could benefit from them. This part is convincing and very well written.

Comment: I have one minor comment: An 68 fold increase of the thermal conductivity is mentioned, which disagrees with the number (42) given in the main text body.

[Authors' Response]: Thank you for pointing this out, this was an oversight. The 68 fold is estimated from measurement data and 42 is from FEM simulations. We have corrected it from 42 to 68 in the text body, and the discussion of simulations is in Supplementary Information.

REVIEWERS' COMMENTS

Reviewer #1 (Remarks to the Author):

My comments on the prior version of this manuscript, as well as comments from the other referees, have all been answered and the manuscript corrected where needed, to my satisfaction. I fully support publication in Nature Communications.

Reviewer #2 (Remarks to the Author):

The authors answered and added discussion in Supplementary Information satisfactory. I recommend publication as it is.

Reviewer #3 (Remarks to the Author):

I have read the rebuttal, revised manuscript and revised supplementary notes related to the submitted work "Two-Dimensional Optomechanical Crystal Cavity With High Quantum Cooperativity" from Prof. Oskar Painter and co-workers. I am globally satisfied by the response to my comments, and can therefore recommend publication without further revision.

P. Verlot